# Dangerous appetite: The impact of *Trypanosoma cruzi* infection on the feeding and defecation behaviors of *Triatoma dimidiata* sensu lato (Latreille, 1811)

Irving Jesús May-Concha[1]*, Víctor Andrés Garrido-González[1], Guadalupe Ivette Hernández-Bolio[1], Mirely del Carmen Franco-Sosa[1], Joel Israel Moo-Millan[1], Etienne Waleckx[1,2,3]*

**1** Laboratorio de Parasitología, Centro de Investigaciones Regionales "Dr. Hideyo Noguchi", Universidad Autónoma de Yucatán, Mérida, México, **2** INTERTRYP, Univ Montpellier, Cirad, IRD, Montpellier, France, **3** ACCyC, Asociación Chagas con Ciencia y Conocimiento, A. C., Orizaba, Veracruz, México

* irving.may@correo.uady.com.mx (IJMC); etienne.waleckx@ird.fr, etienne.waleckx@correo.uady.mx (EW)

## Abstract

### Background

*Trypanosoma cruzi*, the causative agent of Chagas disease, is primarily transmitted through the infected feces of blood-sucking bugs known as triatomines. As a result, its transmission is closely linked to the feeding and defecation behaviors of these insects. *T. cruzi* can alter various physiological processes in its vectors, including those involved in parasite acquisition, development, and excretion. This study aimed to assess the feeding and defecation behaviors of *Triatoma dimidiata* sensu lato (Latreille,1811), one of the main vectors, in relation to its *T. cruzi* infection status.

### Methodology/Principal findings

Using both *T. cruzi*-infected and uninfected *T. dimidiata*, we measured various variables related to feeding and defecation behaviors. Notably, infected insects reached their host four times faster and began defecating twice as quickly as uninfected ones ($282 \pm 58$ s *vs.* $1132 \pm 198$ s; $580 \pm 120$ s *vs.* $1220 \pm 166$ s, respectively), and these differences were significant. Among the bugs that defecated, 89% (16/18) of infected insects and 70% (14/20) of uninfected insects defecated during feeding. Moreover, among the bugs that defecated, a significantly higher proportion of infected insects defecated within the first 10 minutes after the start of feeding (11/18 = 61%) compared to uninfected insects, in which this behavior was less frequent (5/20 = 25%) and occurred later. Additionally, infected insects presented a significantly greater blood meal intake and feeding rate.

**Data availability statement:** All data generated and analyzed during this study is included in the article. Raw data is provided in Supplementary files.

**Funding:** IJMC and EW received financial support from the IRD (French National Research Institute for Sustainable Development) JEAI program (Project ID: ENVY — Ecology of Vector-borne NTDs in Yucatán).The funders had no role in study design, data collection and analysis, decision to publish, or preparation of the manuscript.

**Competing interests:** The authors have declared that no competing interests exist.

## Conclusions/Significance

Overall, these findings suggest that *T. cruzi* alters the feeding and defecation behaviors of *T. dimidiata* in a way that could enhance its transmission potential.

## Author summary

Chagas disease is a major public health concern in Latin America. It is caused by the parasite *Trypanosoma cruzi*, which is primarily transmitted when infected triatomine bugs defecate during or shortly after feeding on a host, depositing the parasite-laden feces on the host's skin. The parasite can then enter the host's organism through mucous membranes or skin lesions. This kind of transmission is known as stectorarian transmission. As such, certain aspects of feeding and defecation behaviors of triatomines, such as the elapsed time between the start of feeding and the start of defecation, play a crucial role in transmission. A vector that defecates on the host during or immediately after feeding increases the likelihood of parasite deposition on the skin, facilitating infection, compared to a vector that defecates after it left the vicinity of the host. Additionally, it is well known that some pathogens can manipulate their vectors by modifying various of their behaviors to enhance transmission likelihood. In this study, we investigated whether *T. cruzi* infection influences the feeding and defecation behaviors of *Triatoma dimidiata*, the major vector of Chagas disease in Southeastern Mexico, in a way that could enhance its transmission potential. Our results show that infected bugs detect their feeding host more quickly, feed more efficiently, and defecate sooner than uninfected bugs. These behavioral changes may enhance parasite transmission, suggesting a potential case of parasite-driven manipulation. Understanding these effects could help refine vector control strategies and improve Chagas disease prevention efforts.

## Introduction

*Trypanosoma cruzi*, the causative agent of Chagas disease in humans, is primarily transmitted by blood-sucking insects of the subfamily Triatominae (Hemiptera: Reduviidae) and circulates among wild, synanthropic, and domestic mammalian hosts [1]. Vectorial transmission occurs when the feces of an infected insect come into contact with mucous membranes or damaged skin of its mammalian blood-feeding host. This typically happens during or immediately after blood feeding, as the insect defecates near the bite site, allowing *T. cruzi* to enter the host's body [2].

Consequently, feeding and defecation behaviors are a critical part of vectorial capacity, as they directly influence the likelihood of *T. cruzi* transmission. The probability of the parasites contained in the feces coming into contact with and infecting a host depends on several parameters of these behaviors, such as feeding duration, number of bites during feeding, timing and number of defecations, and the distance from the host at

which the first post-feeding defecation occurs. These factors define an efficient *T. cruzi* vector and can significantly impact the epidemiology of Chagas disease [3]. Studies on various triatomine species have shown that those exhibiting prolonged feeding behavior, performing a higher number of bites, and defecating during or shortly after blood feeding, particularly while still on or near the host, have an increased likelihood of transmitting *T. cruzi*, thereby enhancing their vectorial capacity [3–9].

Additionally, parasites can modify different physiological, behavioral, and morphological traits of their hosts [10]. These modifications may result from adaptive manipulation by the parasites to enhance their transmission, an adaptive response of the host to the infection, or merely a by-product of the infection that, under certain circumstances, can fortuitously favor parasite transmission [11–13]. In this regard, a growing corpus of studies suggests that *T. cruzi* modifies different characteristics of its vectors, including their behaviors, physiology, and life-history traits [10,11]. Similarly, previous studies indicate that triatomines may experience modifications in their feeding and defecation behaviors when infected with *T. cruzi* [14]. For example, Botto-Mahan et al. [15] found that infected *Mepraia spinolai* detect their blood-feeding host almost twice as fast as uninfected bugs, bite twice more often, and begin defecation earlier. Pereyra et al. [16] reported that *T. cruzi*-infected *T. infestans* defecate twice as fast and in greater quantities than their uninfected counterparts. Chacón et al. [17], in the same species, demonstrated that infected bugs detect their blood-feeding host twice as fast as uninfected bugs, that the number of bites is increased from 4.5 bites in uninfected bugs to 8 bites in infected bugs, and that the time to first defecation is reduced by half in infected bugs. Besides, a significant correlation was found between the parasite load and the behavioral changes observed in infected triatomines [17]. Recently, Killets et al. [18] reported higher defecation index (DI), that the authors defined as DI = (% of insects that defecated up to 10 min post feeding X average number of defecations up to 10 min post feeding)/100, in *T. cruzi*-infected *R. prolixus*, *T. sanguisuga*, and *T. gerstaeckeri* compared to their uninfected counterparts. In the same way, the blood-meal size (volume of blood ingested) and gain weight of *T. cruzi*-infected *R. prolixus* and *T. sanguisuga*, were higher compared to their uninfected counterparts, while for *T. gerstaeckeri,* this depended on the infecting *T. cruzi* strain. In contrast, Takano-Lee and Edman [19] did not find differences between infected and uninfected *R. prolixus* regarding the number of feeding attempts, feeding time, time to the first fecal drop, or number of fecal drops, and D'Alessandro and Mandel [20] reported that *T. cruzi*-infected *Rhodnius prolixus* take blood meals almost twice less frequently as their uninfected counterparts.

*Triatoma dimidiata* (Latreille, 1811) is one of the main vectors of *T. cruzi*, with a wide geographic range extending from Mexico to northern Peru [21]. It is in fact a complex including different genetic lineages [22–24], referred to as *T. dimidiata* sensu lato [23]. Two new cryptic species belonging to this complex have been recently proposed: *T. mopan* [25] and *T. huehuetenanguensis* [26]. Like other triatomines, *T. dimidiata* sensu lato displays diverse behaviors, including, among others, aggregation [27], alarm and defensive responses [28], host-seeking [29], feeding and defecation patterns [30], and mating behaviors [31]. However, research comparing these behaviors between *T. cruzi*-infected and uninfected *T. dimidiata* remains limited. In recent years, our group has been investigating the characteristics of *T. cruzi* infection status in *T. dimidiata* sensu lato. For example, we found that naturally infected *T. dimidiata* have a higher number of antennal receptors compared to uninfected bugs [32]. Additionally, we have observed that *T. cruzi* infection influences both aggregation and host-seeking behaviors, potentially increasing the likelihood of transmission [33,34]. However, there is no previous report comparing the feeding and defecation behaviors of *T. cruzi*-infected and uninfected *T. dimidiata*.

In this study, we evaluated the feeding and defecation behaviors in both infected and uninfected *T. dimidiata* and assessed potential differences associated with *T. cruzi* infection status. We tested the hypothesis that *T. cruzi* infection impacts the feeding and defecation behaviors of *T. dimidiata* in a way that enhances transmission potential.

## Materials and methods

### Ethics statement

The study was conducted in compliance with ethical standards and was approved by the Institutional Bioethics Committee of the Autonomous University of Yucatan, which reviewed and authorized the protocol for animal care and use. Mice and

pigeons used as blood sources for triatomines were handled under Mexican national guidelines (NORMA Oficial Mexicana NOM-062-ZOO-1999, http://www.fmvz.unam.mx/fmvz/ principal/archivos/062ZOO.PDF) for animal care and use. Their care included continuous monitoring by qualified professionals, dietary supplementation with iron and multivitamins, and a scheduled use plan based on insect populations needs. No endangered or protected species were used for this study.

## Insects

Second-instar nymphs of *T. dimidiata* were obtained from a colony maintained at the Parasitology Laboratory of the Regional Research Center Dr. Hideyo Noguchi, Autonomous University of Yucatan. The colony is reared under controlled conditions of 27 ± 1 °C, 70 ± 10% relative humidity (RH), and a 12: 12 h (L: D) photoperiod. The insects are fed on immobilized pigeons (*Columba livia*) between once and three times a week, depending on the experimental requirements of our laboratory. The specimens used in this study came from the fifth laboratory generation of this colony, which originated from *T. dimidiata* individuals collected in rural human dwellings in Yucatan, Southeastern Mexico, and belonging to haplogroup 1 [28], equivalent to ITS-2 group 3 of Bargues et al. [22], later proposed as *T. huehuetenanguensis* by Lima-Cordón et al. [26]. Therefore, any reference to *T. dimidiata* in this manuscript should be interpreted as referring to the species complex *T. dimidiata* sensu lato (Latreille, 1811).

### *Trypanosoma cruzi*

For infection of triatomines, the "H1 strain", a TcI strain of *T. cruzi* (NCBI BioProject accession number PRJNA880352), originally isolated from human and maintained in the laboratory by cyclical passages in BALB/c adult mice, was used.

## Insect infection with *Trypanosoma cruzi*

After a two-week starvation period, the initial infection of the triatomines was carried out in nymphs that had molted to their third instar. The nymphs were fed *ad libitum* on different BALB/c mice (Fig 1A), each previously inoculated with the same *T. cruzi* inoculum. Inoculations were performed 15 days before triatomine feeding, ensuring that all mice were in the exponential growth phase of the parasite according to previous studies carried out in our laboratory [32,35]. Before feeding the nymphs, each mouse was anesthetized following intramuscular injection of a Xylazine–Ketamine mixture (10 mg and 100 mg per kg, respectively). Approximately 30 days after infection, fecal samples were obtained from each bug by gently compressing the terminal abdominal segment with tweezers (Fig 1B). Collected fecal samples were diluted in saline solution containing 0.05% EDTA. The infection status was confirmed by examining the samples under a light microscope at 400 × magnification and observing motile, flagellated parasites. The the number of parasites was quantified using a Neubauer counting chamber. Parasite loads in infected insects ranged from $1.5x10^6$ to $2.5x10^6$ parasites mL$^{-1}$ of feces. The control group was fed under the same conditions using uninfected mice. The nymphs from both groups were then kept under rearing conditions and fed fortnightly on uninfected mice until they molted to the fifth instar. After molting,

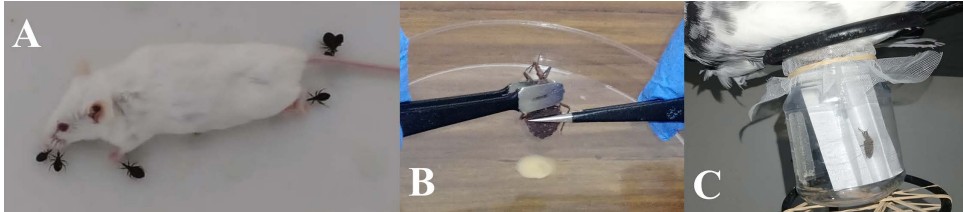

**Fig 1. Key steps of the experimental procedure used in this study (A) Experimental infection of *T. dimidiata* third instar nymphs using anesthetized BALB/c mice.** (B) Collection of feces for parasitemia analysis. (C) Feeding device used for feeding and defecation behavior assessment.

insects were starved for 10–15 days before being used in behavioral experiments. Two experimental groups were established: one consisting of *T. cruzi*-infected insects and the other of uninfected insects. Each experimental group included 20 insects.

### Feeding and defecation patterns

Using a procedure previously reported by Lobbia et al. [36], each triatomine was individually weighed on an analytical balance (±0.0001 g) before being introduced into the experiment. Then, each insect was placed inside a transparent plastic container with a piece of folded paper positioned vertically to facilitate climbing (Fig 1C). After a 5-minute acclimation period, the open side of the container (covered with tulle mesh) was positioned under the ventral side of an immobilized pigeon, allowing the insects to feed *ad libitum* after climbing the folded paper. Time recording started immediately after the acclimation period. After feeding, each insect was carefully placed in a circular glass observation arena with a diameter of 15 cm, lined with filter paper at the bottom, and allowed to walk freely for one hour. Each insect was then weighted again, as well as the feces excreted during the assay. After each observation, the device was cleaned with detergent, chlorine, and alcohol and dried for 30 min to eliminate chemical cues that could affect subsequent assays. All assays were performed in an experimental room in a controlled environment maintained at $23 \pm 1°C$ and $37 \pm 10\%$ relative humidity during the scotophase. The illumination was provided by a 60-W red bulb positioned 120 cm above the arena, with a light intensity of 20 lux. Each pigeon was used for only one experimental session per day, and each triatomine was used only once. The experimenter was blinded to the infection status of each bug to eliminate potential observer bias in the results.

To analyze the feeding and defecation behaviors of *T. dimidiata*, the following previously proposed variables [7,16,17] were recorded for each triatomine: 1) host detection time, defined as the duration from when the triatomine begins to move and walks toward the host until it bites, indicated by the insertion of its rostrum into the pigeon's skin; 2) number of bites, total number of bites performed during feeding; 3) blood meal intake, calculated by subtraction of the initial weight (weight of insect before feeding) from the final weight (the sum of weights of feces and the weight at the end of the assay); 4) feeding duration; 5) feeding rate, calculated by subtraction of the final weight from the initial weight, and dividing by the feeding duration; 6) time to first defecation after the start of feeding; 7) total number of defecations (feces and/or urine) recorded after the start of feeding; 8) proportion of insects that defecated during the assays; 9) proportion of them that defecated during feeding and 10) during the first ten minutes after the start of feeding.

### Statistical analysis

To compare response variables between infected and uninfected groups, we used *t*-tests when the data followed a normal distribution as determined by Shapiro-Wilk tests [37]; otherwise, non-parametric Mann–Whitney *U* tests were applied. The proportions of insects that defecated during the assays, proportions of them that defecated during feeding, and within the first ten minutes after the start of feeding were compared using Fisher's exact tests. All statistical analyses were performed in R 4.4 (Core Development Team, 2020). A significant level of $P < 0.05$ was used.

## Results

### Feeding behavior of *T. dimidiata*

In this section, data are presented as mean ± SEM. Overall, including both infected and uninfected insects, we found a mean host detection time of $707 \pm 123$ s (12 min), a mean number of bites per event of $4.6 \pm 0.4$, a mean blood meal intake of $0.29 \pm 0.03$ gr, a mean feeding duration of $1663 \pm 139$ sec (27.7 min), and a mean feeding rate of 0.2 mg/s $\pm 0.024$ mg/s.

Both the infected and uninfected bug groups showed the same initial weight status before the experimental feeding with an average weight of $0.11 \pm 0.01$ g for both groups. This similarity helped avoid potential confusion related to bug size and fitness.

We observed no significant differences between experimental groups regarding the number of bites (4.0±0.5 *vs.* 5.3±0.5 for infected and uninfected bugs, respectively), or the feeding duration (1642±211 s *vs.* 1684±187 s for infected and uninfected bugs, respectively) (Mann–Whitney U tests, *P*>0.05, Table 1).

Strikingly, infected insects had a host detection time that was four times shorter than that of uninfected insects (282±58 s *vs.* 1132±198, respectively), and this difference was significant (*U*=102, *P*=0.002, Table 1). Additionally, infected insects showed a significantly higher blood meal intake (0.35±0.05 *vs.* 0.22±0.03 g, respectively, t=-2.48, df=38, *P*=0.019) (see Table 1 and Fig 2). Finally, infected insects showed a significantly higher feeding rate (0.26±0.04 mg/s *vs.* 0.13±0.01 mg/s, respectively, U=315, *P*=0.002, Table 1).

## Defecation behavior of *T. dimidiata*

Overall, 38/40 (95%) of the insects defecated during the assays. The difference between groups was not statistically significant (18/20=90% for infected insects *vs.* 20/20=100% for uninfected insects, Fisher's exact test, *P*=0.487, Table 1).

**Table 1. Summary values of the feeding and defecation behaviors according to *T. cruzi* infection status of *T. dimidiata*. Variable values are presented as mean±SEM.**

| Variable | Uninfected insects | Infected insects | Statistical values |
|---|---|---|---|
| Initial weight (grams) | 0.11±0.01 | 0.11±0.01 | *t*=0.04, *df*=38, *P*=0.968 |
| Host detection time (seconds) | 1132±198 | 282±58 | *U*=102, *P*=**0.002** |
| Number of bites per event | 5.3±0.5 | 4.0±0.5 | *U*=146, *P*=0.078 |
| Blood meal intake (grams) | 0.22±0.03 | 0.35±0.05 | *t*=-2.48, *df*=38, *P*=**0.019** |
| Feeding duration (seconds) | 1684±187 | 1642±211 | *U*=186, *P*=0.598 |
| Feeding rate (mg/s) | 0.13±0.01 | 0.26±0.04 | *U*=315, *P*=**0.002** |
| Time to first defecation after the start of feeding (s) | 1220±166 | 580±121 | *U*=274, *P*=**0.006** |
| Number of defecations | 5.0±0.6 | 3.9±0.5 | *t*=-1.54, df=38, *P*=0.13 |
| Proportion of insects that defecated during the assays (%) | 100 | 90 | Fisher's exact test, *P*=0.487 |
| Proportion of insects that defecated during feeding (among those that defecated during the assays) (%) | 70 | 89 | Fisher's exact test, *P*=0.238 |
| Proportion of insects that defecated during the first ten minutes after the start of feeding (among those that defecated during the assays) (%) | 25 | 61 | Fisher's exact test, *P*=**0.047** |

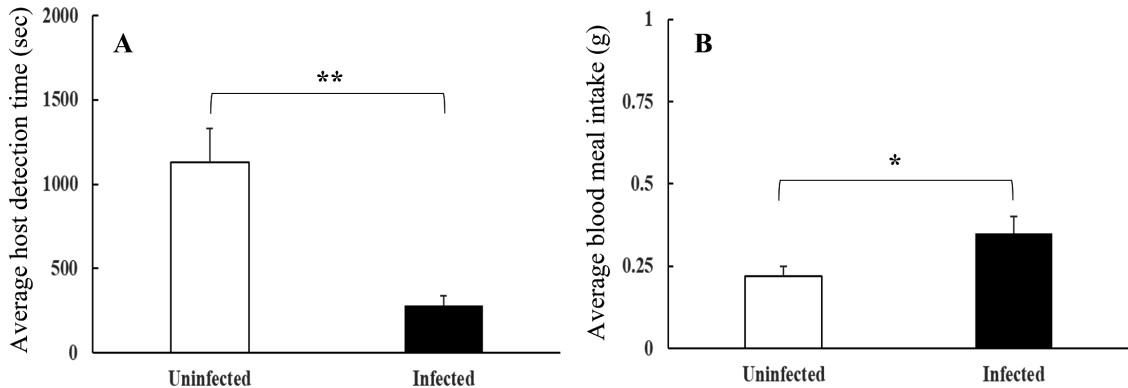

**Fig 2. Feeding behavior according to *Trypanosoma cruzi* infection status of *Triatoma dimidiata*.** (A) Host detection time, in seconds, for *T. cruzi*-infected and uninfected *T. dimidiata*. (B) Blood meal intake, in grams, for *T. cruzi*-infected and uninfected *T. dimidiata*. Significant differences between paired bars (Kruskall-Wallis test) are indicated by *\*P*<0.05, *\*\*P*<0.01. Error bars represent the standard error.

The insects that defecated during the assays began defecating between five and 2555 seconds after the start of feeding. Of them, infected bugs began defecating twice as fast as uninfected ones (580 ± 121 seg *vs.* 1220 ± 166 s, respectively, see Table 1 and Fig 3), and this difference was significant (*U* = 274, *P* = 0.006). However, no significant difference was observed in the total number of defecations when considering all the bugs used in the assays (3.9 ± 0.5 *vs.* 5.0 ± 0.6 for infected and uninfected bugs, respectively, *t* = -1.54, df = 38, *P* = 0.13, Table 1), or considering only the bugs that defecated during the assays (4.5 ± 0.5 *vs.* 5.0 ± 0.6 for infected and uninfected bugs, respectively, *t* = -1.01, df = 36, *P* = 0.32).

Among the insects that defecated during the assays, 16/18 (89%) of infected insects and 14/20 (70%) of uninfected insects defecated while feeding. Although this difference was not statistically significant (Fisher's exact test, *P* = 0.238), the proportion that defecated within the first 10 minutes after the start of feeding was significantly higher in infected insects (11/18 = 61%) than in uninfected ones (5/20 = 25%) (Fisher's exact test, *P* = 0.047, Table 1).

## Discussion

In this study, we evaluated the feeding and defecation behaviors of *T. dimidiata* according to *T. cruzi* infection status, as these behaviors are key determinants of vectorial capacity and the likelihood of *T. cruzi* transmission [38]. Our study supports the hypothesis that *T. cruzi* infection alters the feeding and defecation behaviors of *T. dimidiata* in a way that increases the likelihood of parasite transmission, potentially as a result of vector manipulation by *T. cruzi,* as suggested in previous studies [15,39].

Firstly, we observed that infected *T. dimidiata* reached their blood-feeding host four times faster than uninfected bugs. A similar reduction in host detection time has been reported in infected *M. spinolai* and *T. infestans* [15,17,40], with infected individuals locating their hosts nearly twice as fast as uninfected ones [15,17]. These findings suggest that infection may influence the sensory biology and the host-seeking behavior in triatomines, including *T. dimidiata* [41,42]. Additionally, the increased number of certain antennal receptors in infected *T. dimidiata*, as previously reported [32], may enhance their ability to detect/identify potential blood-feeding hosts.

Secondly, while two infected bugs did not defecate during the assays, we did not find statistical difference in the proportion of bugs that defecated between the two groups (90% for infected bugs *vs.* 100% for uninfected bugs), and we found that, among those that did defecate, infected *T. dimidiata* began defecating twice as fast as uninfected bugs. Moreover,

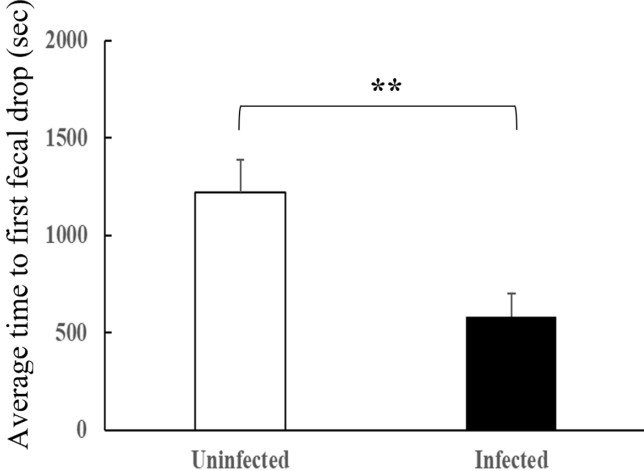

**Fig 3. Defecation behavior according to *Trypanosoma cruzi* infection status of *Triatoma dimidiata*.** Elapsed time between the start of feeding and the start of defecation, in seconds, for *T. cruzi*-infected and uninfected *T. dimidiata*. Significant differences between paired bars (Kruskall-Wallis test) are indicated by **P* < 0.05, ***P* < 0.01. Error bars represent the standard error.

among the bugs that defecated, although the proportion that defecated during feeding was not significantly different between groups (80% for infected bugs *vs.* 70% for uninfected bugs), the proportion that defecated within the first 10 minutes after the start of feeding was significantly higher in infected bugs (61.1% *vs.* 25.0%). This finding has important implications, as the most efficient vectors are generally those that defecate during or immediately after feeding, thereby increasing the likelihood of infective feces coming into contact with the host [4,5,43,44].

Third, we found that the blood meal intake and the feeding rate were significantly higher in infected *T. dimidiata* compared to uninfected bugs. This finding aligns with previous studies on *M. spinolai*, *T. rubrovaria, T, sanguisuga,* and *T. gerstaeckeri* [15,18,45] and suggests that *T. cruzi* infection positively influences these feeding parameters. As previously proposed, a possible explanation is the competition for the nutrients in the ingested blood between the triatomine and *T. cruzi* [17].

Besides the differences observed between *T. cruzi*-infected and uninfected *T. dimidiata*, some measured parameters showed no significant differences between groups. Specifically, we found no differences in the number of bites or feeding duration. Our findings are consistent with those of Pereyra et al. [16] in *T. infestans* but differ from previous studies reporting that *T. cruzi*-infected *R. prolixus*, *T. rubrovaria* and *T. infestans* show changes in the number of bites and feeding duration [17,20,45]. One possible explanation is that fasting before the experiment influenced these results, as suggested by Pereyra et al. [16]. Alternatively, it is possible that *T. cruzi* infection does not affect these specific parameters in *T. dimidiata*.

Among the limitations of our study is the small sample size (n = 20 per group), which reduced the statistical power of our analyses, and may limit the generalizability of our findings. Nevertheless, our sample size was comparable to or even larger than those used in previous research investigating feeding and defecation patterns in triatomines [7,19,46]. Another factor that may limit the generalizability of our results is that we tested only a single *T. cruzi* strain, whereas behavior effects may vary depending on the strain [18]. Additionally, the range of parasite loads in the insects could represent a potential limitation. However, in our study, parasitemia was relatively homogeneous across all insects, ranging from 1.5 to 2.5 x $10^6$ parasites mL$^{-1}$ of feces. Although variation in parasite load can influence vector behavior, evidence from Chacón et al. [17] suggests that only large differences in infection intensity are likely to cause significant behavioral changes, indicating that the range observed here, remaining within the same order of magnitude, is unlikely to have biased our results. Finally, the variable starvation time experienced by the insects, which ranged from 10 to 15 days, could have influenced individual host-seeking and feeding behaviors. However, variable starvation time between bugs is a common feature of experimental studies with triatomines, as their highly variable molting times make it challenging to standardize the starvation period precisely before their use in experiments [16]. Moreover, previous studies indicate that starvation durations within the range of 10–15 days have no significant impact on the feeding responses of triatomines [47]. Therefore, it is reasonable to conclude that this range of starvation time is suitable for assessing differences in feeding behavior without introducing significant variability. Despite these limitations, our study lays the groundwork for further research to investigate, for example, the influence of parasite strain, parasite load, and starvation time on the feeding and defecation behaviors of *T. dimidiata* sensu lato.

## Conclusions

The current work demonstrates that the TcI strain of *T. cruzi* used in this study modifies the feeding and defecation patterns of *T. dimidiata*, supporting the hypothesis that *T. cruzi* manipulates *T. dimidiata* to enhance its transmission potential. These findings have significant implications for transmission dynamics and should be considered to enhance understanding of the eco-epidemiology of *T. cruzi* infection and Chagas disease.

## Supporting information

**S1 Table. The complete dataset produced and analyzed in this study.**
(XLSX)

## Acknowledgments

We thank the Secretaría de Ciencia, Humanidades, Tecnología e Innovación (SECIHTI) for support to IJMC/ CVU: 272733. We also extend our gratitude to Pedro Pablo Martínez-Vega and Víctor Manuel Dzul-Huchim for their assistance with mouse handling.

## Author contributions

**Conceptualization:** Irving Jesús May-Concha.

**Data curation:** Irving Jesús May-Concha, Víctor Andrés Garrido-González.

**Formal analysis:** Irving Jesús May-Concha, Víctor Andrés Garrido-González, Joel Israel Moo-Millan, Etienne Waleckx.

**Funding acquisition:** Etienne Waleckx.

**Investigation:** Irving Jesús May-Concha, Víctor Andrés Garrido-González.

**Methodology:** Irving Jesús May-Concha, Víctor Andrés Garrido-González, Guadalupe Ivette Hernández-Bolio, Mirely del Carmen Franco Sosa.

**Project administration:** Etienne Waleckx.

**Resources:** Etienne Waleckx.

**Software:** Irving Jesús May-Concha.

**Supervision:** Irving Jesús May-Concha.

**Validation:** Irving Jesús May-Concha.

**Visualization:** Irving Jesús May-Concha.

**Writing – original draft:** Irving Jesús May-Concha.

**Writing – review & editing:** Irving Jesús May-Concha, Víctor Andrés Garrido-González, Guadalupe Ivette Hernández-Bolio, Mirely del Carmen Franco Sosa, Joel Israel Moo-Millan, Etienne Waleckx.

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
