## [Decision Letter · Decision Letter 0]

12 Aug 2025

Response to Reviewers
Revised Manuscript with Track Changes
Manuscript

Shaden Kamhawi

co-Editor-in-Chief

Paul Brindley

co-Editor-in-Chief

**Journal Requirements:**

At this stage, the following Authors/Authors require contributions: Irving Jesús May-Concha, Víctor Andrés Garrido-González, Ivette Guadalupe Hernández-Bolio, Mirely del Carmen Franco Sosa, Joel Israel Moo-Millan, and Etienne Waleckx. Please ensure that the full contributions of each author are acknowledged in the "Add/Edit/Remove Authors" section of our submission form.

2) We noticed that you used the phrase 'unpublished data' in the manuscript. We do not allow these references, as the PLOS data access policy requires that all data be either published with the manuscript or made available in a publicly accessible database. Please amend the supplementary material to include the referenced data or remove the references.

3) In the online submission form, you indicated that "Raw data is available without restriction under request to the corresponding authors.". All PLOS journals now require all data underlying the findings described in their manuscript to be freely available to other researchers, either

- In a public repository

- Within the manuscript itself

- Uploaded as supplementary information.

4) Please ensure that the funders and grant numbers match between the Financial Disclosure field and the Funding Information tab in your submission form. Note that the funders must be provided in the same order in both places as well.

**Reviewers' comments:**

**Key Review Criteria Required for Acceptance?**

**Methods:**

-Are the objectives of the study clearly articulated with a clear testable hypothesis stated?

-Is the study design appropriate to address the stated objectives?

-Is the population clearly described and appropriate for the hypothesis being tested?

-Is the sample size sufficient to ensure adequate power to address the hypothesis being tested?

-Were correct statistical analysis used to support conclusions?

-Are there concerns about ethical or regulatory requirements being met?

Reviewer #1: This is a well structured and defined manuscript looking at the behaviors patters of infected T. dimidiate Vs non-infected. Authors should stated in the meteorology what was the source of feeding and how the host was managed. (is the¡ere a bioethical permission for this?).

Reviewer #2: 1. The manuscript does not provide essential information regarding the origin and maintenance of the T. dimidiata colony used in the experiments. Given that T. dimidiata is a species complex with significant genetic and behavioral variation across its geographic distribution, it is critical to specify the collection site, the estimated number of laboratory generations, and the approximate number of founding individuals used to establish the colony. These factors may directly influence the behavioral traits observed and are important for evaluating the ecological relevance of the results.

2. Although the authors acknowledge the limited sample size (n = 20), key methodological details concerning the infection protocol are missing. It is not specified whether infected insects were fed on the same or on different mice, whether host parasitemia was measured or standardized, or whether infection intensity in the insects was quantified. These omissions, particularly in conjunction with the small sample size, may introduce variability and potential bias into the behavioral outcomes. The sample size should be treated as a factor limiting the strength of interpretation, rather than justified by precedent.

3. The starvation period prior to the behavioral assays ranged from 10 to 15 days. This 5-day variation is biologically meaningful for triatomines—especially in nymphal stages—as hunger level can significantly influence host-seeking and feeding behavior. In combination with the small sample size, this variation could affect the consistency of the behavioral measurements. While it may not invalidate the results, it should be acknowledged as a potential limitation in the discussion.

Reviewer #3: The objectives were achieved and corresponded to an adequate hypothesis.

Even though the authors support the low number (20) of individuals examined, I believe that a larger number would have shown more reliable results.

Statistical analysis supported the results.

**Results:**

-Does the analysis presented match the analysis plan?

-Are the results clearly and completely presented?

-Are the figures (Tables, Images) of sufficient quality for clarity?

Reviewer #1: Results are clear but should briefly describe if possible the degree of ing¡fiction of the vectors......low, medium high? to have a more clear panorama of the situation.

Reviewer #2: While clearly presented, may be biased due to methodological limitations outlined in the Methods section.

Reviewer #3: They found notable differences in the time required to detect the host in individuals infected with T. dimidiata (minor) compared to uninfected individuals. Infected individuals bit the host more times and defecated in less than 10 minutes compared to uninfected individuals. More infected individuals defecated within 10 minutes of eating than uninfected individuals. In contrast, similar percentages of both infected and uninfected triatomines defecated during feeding. A similar pattern was observed in feeding times.

The figures and tables are appropriate and presented in a clear and satisfactory manner.

**Conclusions:**

-Are the conclusions supported by the data presented?

-Are the limitations of analysis clearly described?

-Do the authors discuss how these data can be helpful to advance our understanding of the topic under study?

-Is public health relevance addressed?

Reviewer #1: Conclusion id according the data presented

Reviewer #2: The manuscript would benefit from a more cautious interpretation of results and a clearer discussion of how these preliminary observations could contribute to broader understanding or public health implications.

Reviewer #3: The data supported the conclusions reached. The authors clearly described the limitations of their work.

Public health could benefit from a better understanding of the behavior of each triatomine species.

**Editorial and Data Presentation Modifications?**

Reviewer #1: Probable th inclusion of pictures of wedding bugs will give a more visual impact o the results.

Reviewer #2: (No Response)

Reviewer #3: Lines 85-88. “The probability of the parasites contained into the feces coming into contact with and infecting a host depends on several parameters of these behaviors, such as feeding duration, number of bites during feeding, and the timing and number of fecal drops. These define an efficient T. cruzi vector and can significantly impact the epidemiology of Chagas disease [3].” The distance from the host of first defecation post feeding likely influences also the chances of infection. Please include that information.

Nogueda-Torres B. et al. (2025). Period of time and movement distances between feeding and postfeeding defecation in Triatoma pallidipennis (Heteroptera: Reduviidae). Acta Trop. 263:107563. doi: 10.1016/j.actatropica.2025.107563.

Lines 89-93. “Studies on various triatomine species have shown that those exhibiting a higher number of bites, defecating during or shortly after blood feeding, and engaging in prolonged feeding behavior increase the likelihood of T. cruzi transmission, thereby enhancing their vectorial capacity [3–7]. The authors cite two articles on T. infestans, two on T. sordida, and one on T. carrioni. Consider diversifying the species to cite and including some Mexican species, such as the one in their study. These studies are abundant and easily accessible.

Lines 102-115. The authors mention the results on various parameters from different authors for various triatomine species, highlighting the differences in favor of infected triatomines compared to uninfected ones. However, it is necessary to include those data (as the authors did in later lines), but even more important is to include those authors' conclusions based on those differences, because in some of them the recorded differences do not convert an ineffective vector into an effective one.

Lines 117-122. Studies in which no differences were found between the groups studied should be explained in more detail, similar to those in which differences were observed.

The Introduction would benefit if the authors read section 3.3 (Effects of T. cruzi on triatomines) of the chapter:

Guarneri A. A. and G. A. Schaub. 2021. Interaction of triatomines with their bacterial microbiota and trypanosomes. (p. 355-363). In: A. Guarneri and M. Lorenzo (Eds.) Triatominae. The Biology of Chagas Disease Vectors. Springer.

Line 123. “Triatoma dimidiata (Latreille)”. Please indicate whether your population belongs to Triatoma dimidiata sensu lato or sensu stricto. Remember to use this name throughout your manuscript.

Lines 124-126 “Like other triatomines, it displays diverse behaviors, including, among others, aggregation, alarm responses, defensive actions, host-seeking, feeding, defecation, and mating behaviors”. It is inappropriate to support all this information with a limited reference. Dr. Zeledón's studies should be included, as should those related to T. dimidiata from Mexico, including some from the UADY group.

Line 342. Modify Triatoma sórdida

**Summary and General Comments:**

Reviewer #1: In the discussion section just to mention that the degree I¡of infection of vectors was not assessed and this may have some effects of the results within the experimental group

Reviewer #2: This manuscript examines the influence of Trypanosoma cruzi infection on the feeding and defecation behaviors of Triatoma dimidiata, an important vector of Chagas disease. While the experimental approach is valid and the topic relevant, the study presents important limitations. In particular, missing methodological details and the restricted scope of the experimental design reduce the interpretability and generalizability of the findings.

Reviewer #3: This is an interesting manuscript from the group of young researchers at UADY linked to the French IRPD group of Dr. Waleckx. As the authors state, this is another of several studies that attempt to clarify whether T. cruzi infection sufficiently modifies the behavior of a triatomine species (T. dimidiata in this case) to increase the chances of the parasite infecting a host. The authors examined the feeding and defecation behaviors of infected and uninfected specimens of T. dimidiata, one of the three most epidemiologically important vectors of Trypanosoma cruzi to human populations in Mexico.

I believe this manuscript provides interesting data to unravel the mystery of the influence of T. cruzi on the behavior of each triatomine species. Therefore, I believe it deserves to be published in PLoS, but before that happens, some basic questions need to be resolved.

PLOS authors have the option to publish the peer review history of their article (what does this mean? ). If published, this will include your full peer review and any attached files.

**Do you want your identity to be public for this peer review?** For information about this choice, including consent withdrawal, please see our Privacy Policy .

Reviewer #1: **Yes: ** Antonio Ortega-Pacheco

Reviewer #2: No

Reviewer #3: No

**Figure resubmission:****Reproducibility:** To enhance the reproducibility of your results, we recommend that authors of applicable studies deposit laboratory protocols in protocols.io, where a protocol can be assigned its own identifier (DOI) such that it can be cited independently in the future. Additionally, PLOS ONE offers an option to publish peer-reviewed clinical study protocols. Read more information on sharing protocols at https://plos.org/protocols?utm_medium=editorial-email&utm_source=authorletters&utm_campaign=protocols

---

## [Decision Letter · Decision Letter 1]

11 Dec 2025

Response to Reviewers
Revised Manuscript with Track Changes
Manuscript

Shaden Kamhawi

co-Editor-in-Chief

Paul Brindley

co-Editor-in-Chief

**Reviewers' comments:**

**Key Review Criteria Required for Acceptance?**

**Methods**

-Are the objectives of the study clearly articulated with a clear testable hypothesis stated?

-Is the study design appropriate to address the stated objectives?

-Is the population clearly described and appropriate for the hypothesis being tested?

-Is the sample size sufficient to ensure adequate power to address the hypothesis being tested?

-Were correct statistical analysis used to support conclusions?

-Are there concerns about ethical or regulatory requirements being met?

Reviewer #1: The methodology is clearly described. Only wondering if authors look for T. Cruzi forms o DNA from the feces of insects after feeding. In Figure 1, was the mice sedated? more information is required.

Reviewer #2: Since the limitations of the manuscript were acknowledged, I think it is now suitable.

Reviewer #3: The objectives are clear and aligned with the hypothesis.

The design is appropriate and leads to the achievement of the objective.

**Results**

-Does the analysis presented match the analysis plan?

-Are the results clearly and completely presented?

-Are the figures (Tables, Images) of sufficient quality for clarity?

Reviewer #1: Only wondering if authors look for T. Cruzi forms o DNA from the feces of insects after feeding.

Reviewer #2: "yes" to all

Reviewer #3: The analysis is correct. The results are presented correctly, and the figures and tables help to better understand the study.

**Conclusions**

-Are the conclusions supported by the data presented?

-Are the limitations of analysis clearly described?

-Do the authors discuss how these data can be helpful to advance our understanding of the topic under study?

-Is public health relevance addressed?

Reviewer #1: Conclusions are clear and concise.

Reviewer #2: "yes" to all.

Reviewer #3: The data presented support the conclusion, and the limitations of the study are mentioned. The relevance of the study to public health is discussed. The Discussion is useful and clear.

**Editorial and Data Presentation Modifications?**

Reviewer #1: Indicate details of figure 1 as stated above

Reviewer #2: Accept

Reviewer #3: The minor modifications previously suggested were followed.

**Summary and General Comments**

Reviewer #1: This is an interesting and well written manuscript. Considering changes made as reviewers requested; now it may be considered for publication

Reviewer #2: Since the limitations of the manuscript were acknowledged, I think it is now suitable.

Reviewer #3: The minor modifications previously suggested were followed, so I believe this manuscript could be published in its current form.

PLOS authors have the option to publish the peer review history of their article (what does this mean? ). If published, this will include your full peer review and any attached files.

**Do you want your identity to be public for this peer review?** For information about this choice, including consent withdrawal, please see our Privacy Policy .

Reviewer #1: **Yes: ** Antonio Ortega Pacheco

Reviewer #2: No

Reviewer #3: No

**Figure resubmission:**

**Reproducibility:** To enhance the reproducibility of your results, we recommend that authors of applicable studies deposit laboratory protocols in protocols.io, where a protocol can be assigned its own identifier (DOI) such that it can be cited independently in the future. Additionally, PLOS ONE offers an option to publish peer-reviewed clinical study protocols. Read more information on sharing protocols at https://plos.org/protocols?utm_medium=editorial-email&utm_source=authorletters&utm_campaign=protocols

---

## [Editor Report · Decision Letter 2]

18 Dec 2025

Dear Dr Waleckx,

We are pleased to inform you that your manuscript 'Dangerous appetite: The impact of Trypanosoma cruzi infection on the feeding and defecation behaviors of Triatoma dimidiata sensu lato (Latreille, 1811).' has been provisionally accepted for publication in PLOS Neglected Tropical Diseases.

Best regards,

Adly M.M. Abd-Alla, Prof asso.

Section Editor

Adly Abd-Alla

Section Editor

Shaden Kamhawi

co-Editor-in-Chief

Paul Brindley

co-Editor-in-Chief

---

## [Editor Report · Acceptance letter]

Dear Dr Waleckx,

We are delighted to inform you that your manuscript, "Dangerous appetite: The impact of Trypanosoma cruzi infection on the feeding and defecation behaviors of Triatoma dimidiata sensu lato (Latreille, 1811).," has been formally accepted for publication in PLOS Neglected Tropical Diseases.

Best regards,

Shaden Kamhawi

co-Editor-in-Chief

Paul Brindley

co-Editor-in-Chief
